

# Identification of a necroptosis-related gene signature for making clinical predictions of the survival of patients with lung adenocarcinoma

Xiaoping Zhou[1],[*], Ming Zhao[2],[*], Yingzi Fan[1] and Ying Xu[1]

[1] Department of Laboratory Medicine, Clinical Medical College and the First Affiliated Hospital of Chengdu Medical College, Chengdu, Sichuan, China
[2] Department of Gastroenterology, Clinical Medical College and the First Affiliated Hospital of Chengdu Medical College, Chengdu, Sichuan, China
[*] These authors contributed equally to this work.

Corresponding authors
Yingzi Fan, 506633834@qq.com
Ying Xu, yingxu825@126.com

## ABSTRACT

**Background:** Lung adenocarcinoma (LUAD) is a major pathological subtype of malignant lung cancer with a poor prognosis. Necroptosis is a caspase-independent programmed cell death mode that plays a pivotal role in cancer oncogenesis and metastasis. Here, we explore the prognostic values of different necroptosis-related genes (NRGs) in LUAD.

**Methods:** mRNA expression data and related clinical information for LUAD samples were obtained from The Cancer Genome Atlas (TCGA) and Gene Expression Omnibus databases. NRGs were identified using the GeneCards database. Least absolute shrinkage and selection operator Cox regression and multivariate Cox analysis were used to construct a prognostic risk model. Time-dependent receiver-operating characteristic curves and a nomogram were constructed to validate the predictive values of the prognostic signatures. A necroptosis-related protein–protein interaction network was visualised using the STRING database and Cytoscape software. Functional analyses, including Gene Ontology, Kyoto Encyclopaedia of Genes and Genomes pathway enrichment, gene set enrichment, and gene set variation analyses, were conducted to explore the underlying molecular mechanisms. Finally, the mRNA expression of the prognostic signatures in LUAD cell lines was assessed using reverse transcription-quantitative polymerase chain reaction (RT-qPCR) analysis.

**Results:** A prognostic model was established for eight NRGs (CALM1, DDX17, FPR1, OGT, PGLYRP1, PRDX1, TUFM, and CPSF3) based on TCGA-cohort data and validated with the GSE68465 cohort. Patients with low-risk scores had better survival outcomes than those with high-risk scores ($p = 0.00013$). The nomogram was used to predict the prognosis of patients with LUAD. The prediction curves for 1-, 3-, and 5-year OS showed good predictive performance and the accuracy of the nomograms increased over time. RT-qPCR results demonstrated that these eight genes, especially CALM1, PRDX1, and PGLYRP1, were differentially expressed in LUAD cells.

**Conclusion:** We constructed a reliable eight-NRG signature that provides new insights for guiding clinical practice in the prognosis and treatment of LUAD.

# INTRODUCTION

Lung cancer remains the leading cause of cancer-related deaths globally, with 1.8 million related deaths occurring per year (*Sung et al., 2021*). Lung adenocarcinoma (LUAD) is the major pathological subtype of lung cancer, accounting for approximately 40% of all lung cancer cases (*Zappa & Mousa, 2016*). For patients with early LUAD, surgery with chemoradiotherapy can significantly improve the 5-year survival rate, but patients who present with advanced LUAD have very poor 5-year survival rates because of metastatic lesions. Therefore, the identification of novel biomarkers that predict survival probability and provide new therapeutic targets for LUAD is an urgent priority.

Necroptosis, a caspase-independent mode of programmed cell death that is mainly regulated by the core RIPK3 and MLKL proteins (*Cai & Liu, 2014*), is characterised by cell-membrane rupture, cytoplasmic adenosine triphosphate degradation, and the release of damage-related molecular modules, cytokines, and chemokines. Necroptosis plays a dual role in cancer, inhibiting and promoting oncogenesis and cancer progression, with its specific role often depending on the tumour type and developmental stage. For example, RIPK3 was not expressed in two-thirds of over 60 cancer cell lines examined in a previous study (*Gong et al., 2019*), suggesting that tumours acquire resistance to necroptosis to survive. However, necroptosis can also promote the genesis, development, invasion, and metastasis of tumours by stimulating inflammatory reactions in the tumour microenvironment. Moreover, many key molecules involved in necroptosis have been identified as potential predictors of the overall survival (OS) of patients. Low RIPK3 expression is associated with a reduced OS in patients with colorectal (*Feng et al., 2015*) and breast (*Stoll et al., 2017*) cancer. Low MLKL expression levels have been detected in multiple cancer cell lines and several cancer types. Decreased MLKL expression is associated with a poor prognosis for patients with gastric (*Ertao et al., 2016*), ovarian (*He et al., 2013*) and colon (*Li et al., 2017*) cancer. Conversely, high levels of phosphorylated MLKL were associated with decreased OS in patients with colon and oesophageal cancer (*Liu et al., 2016*). The cylindromatosis protein is a key molecule that mediates necroptosis in chronic lymphocytic leukaemia, and its downregulation is often indicative of a worse OS for patients (*Wu et al., 2014*). However, data on whether necroptosis-related genes (NRGs) are associated with the prognosis of LUAD are limited.

In this study, we identified prognostic NRG markers for LUAD and performed functional enrichment analysis for the resulting gene set. We successfully constructed both risk and prediction models using a training cohort, obtained from The Cancer Genome Atlas (TCGA). Furthermore, the models were supported by verification using an external validation cohort from Gene Expression Omnibus (GEO). We determined the expression of select genes using reverse transcription-quantitative polymerase chain reaction (RT-qPCR) analysis and utilised the clinical information of patients to establish a nomogram for calculating their survival probabilities. The results of this study provide novel insights into the prognosis and clinical management of patients with LUAD.
## MATERIALS AND METHODS

### Data acquisition

We downloaded the expression matrix of the LUAD dataset (TCGA-LUAD) from TCGA using the R package TCGAbiolinks (*Colaprico et al., 2016*) (https://portal.gdc.cancer.gov/). We obtained the information on 523 LUAD samples (cancer group) and 55 paracancerous samples (normal group). After excluding samples with incomplete clinical information, RNA-seq data for 498 samples with complete clinical information were used as the training dataset. All samples were included in this study and standardised to the fragments per kilobase million format, while the corresponding clinical data were obtained from the UCSC Xena database (*Goldman et al., 2020*) (http://genome.ucsc.edu). The count sequencing data of the dataset (TCGA-LUAD) were standardised using the R package limma (version 3.56.1). We also downloaded the LUAD dataset, GSE68465 from the GEO database (*Barrett et al., 2013*) using the R package GEOquery (*Davis & Meltzer, 2007*). All samples of GSE68465 were from *Homo sapiens*. The platform for data generation was GPL96 [HG-U133A] Affinemetrix Human Genome U133A Array. GSE68465 includes microarray gene expression profiles of 442 patient samples of LUAD and 20 normal samples. The dataset probe name annotation uses the chip GPL platform file. All samples were included in this study. Subsequently, 371 samples with prognostic information were included in riskscore validation analysis. Immunohistochemical data were derived from the Human Protein Atlas (HPA) database (https://www.proteinatlas.org/).

### Constructing a risk and clinical prognostic model

To study the effect of necroptosis on the prognosis of patients with LUAD, we searched the GeneCards database (https://www.genecards.org) for the keyword 'necroptosis' and obtained 614 NRGs (Table S1). We also obtained an expression matrix for NRGs by combining TCGA data in our analysis and used the least absolute shrinkage and selection operator (LASSO) Cox regression to identify key genes related to the prognosis of LUAD patients. Subsequently, we included the characteristic genes related to LUAD into the regression model and acquired hazard ratio and 95% confidence interval data by performing multivariate logistic regression. To construct a clinical prediction nomogram, we used the R package rms to develop a risk score and characteristic gene model.
To quantitatively assess the differentiation performance of the nomogram, we generated a calibration curve and used it to compare the values predicted by the nomogram with the observed survival rate.

### Biological function analysis

The Gene Ontology (GO) and Kyoto Encyclopedia of Genes and Genomes (KEGG) pathway analysis of NRGs were performed through the R 'clusterProfiler' package; gene set enrichment analysis (GSEA) analysis was conducted with the same method. Data were obtained from 'c2.kegg.v7.4.symbols' and 'c5.go.v7.4.symbols' gene sets in the MSigDB database. The false-discovery rate and an adjusted *P* value (*P*.adjust) of <0.05 were used as screening criteria.

**Table 1 Primer sequences of hub genes.**

| Prime name | Sequence |
| --- | --- |
| h-CALM1-F | CGCTGCTGTGTCTCGTC |
| h-CALM1-R | AGTATGCTGAGGGGTTCGT |
| h-TUFM-F | CTGAGATGGTGGAACTGGTGGAAC |
| h-TUFM-R | ACAGAGAGCAGAGCCTACGATGAC |
| h-CPSF3-F | ACGTGAAGAGCGAGAAGCAAGATTC |
| h-CPSF3-R | CCTGAGCCCTTCCAAGAGCAAAG |
| h-FPR1-F | GCTGTATCTGCTGGCTATCTCTTCC |
| h-FPR1-R | GGTAACTGATGGTGGTGACTGTGTG |
| h-OGT-F | GTACGGGTTACCAGAAGATGCCATC |
| h-OGT-R | ACGCAACAGCCAGAGTACACTATTG |
| h-PRDX1-F | TGGTGCTTCTGTGGATTCTCACTTC |
| h-PRDX1-R | CGCTTCGGGTCTGATACCAAAGG |
| h-DDX17-F | CGTGGGCTAGATGTGGAAGATGTC |
| h-DDX17-R | TTGTTGGTGCTACGGGCTGTTC |
| h-PGLYRP1-F | AGCGGCTCAGGAGACAGAAGAC |
| h-PGLYRP1-R | ATAGCGTAAGGGCAGGCTCAGG |

Gene Set Variation Analysis (GSVA) was used to explore the molecular and biological differences with the R 'GSVA' package. The 'c5.all.v7.4.symbols' file was used as the reference gene set. $P$.adjust < 0.05 were considered significant enrichment.

## Constructing a protein–protein-interaction network

A protein-protein interaction (PPI) network of proteins encoded by NRGs was analysed using the STRING database (https://string-db.org/). We used the MCODE plug-in unit of the Cytoscape software (version: 3.9.1) to extract the PPI subnet. The PPI network was also visualised using the Cytoscape software.

## Cell culture

A human normal lung cell line (2B) and two LUAD cell lines (A549 and H1650) were purchased from Beyotime Biotechnology (Procell, Wuhan, China) and cultured in RPMI-1640 medium containing 10% foetal bovine serum at 37 °C in 5% $CO_2$. All media and supplements were purchased from Invitrogen (Carlsbad, CA, USA).

## Total RNA extraction and RT-qPCR analysis

Total RNA was extracted from three cell lines (2B, A549, and H1650) using the Total RNA Extraction Kit (Solarbio, Beijing, China). RNA was reverse-transcribed into complementary DNA using the PrimeScript™ RT Reagent Kit (Bio-Rad, Hercules, CA, USA). The mRNA level of hub genes was determined with RT-qPCR using SYBR Green Supermix (Bio-Rad, Hercules, CA, USA) on a CFX96 real-time system (Bio-Rad, Hercules, CA, USA). The sequences of the synthesised primers (Shenggong, Shanghai, China) are listed in Table 1.

## Statistical analyses

The statistical software package, R (version 4.0.2), was utilised for all statistical analyses. When comparing two groups with continuous variables, we used the Student's *t*-test and the Mann–Whitney *U* test to analyse data with a normal distribution or a abnormal distribution, respectively. The pROC R package was used to generate receiver-operating characteristic (ROC) curves and calculate the area under the curve (AUC) to evaluate the accuracy of the risk score in estimating the prognosis. *P* < 0.05 was considered to reflect a statistically significant difference.

## RESULTS

### Construction of the necroptosis prognosis signature

Our study design is represented by the flowchart depicted in Fig. 1. We analysed RNA-seq data from 498 primary LUAD samples in TCGA to identify the prognosis-related genes. Subsequently, 614 genes were obtained from the GeneCards database with the term 'necroptosis'. We built a proportional risk model based on the NRGs with *P* < 0.05 as the screening criterion (Figs. 2A and 2B). Univariate and multivariate Cox regression analyses were performed to screen for independent risk factors of necroptosis in LUAD, which identified Calmodulin 1 (CALM1), DEAD-box RNA helicase 17 (DDX17), formyl peptide receptor 1 (FPR1), O-GlcNAc transferase (OGT), peptidoglycan recognition protein (PGLYRP1), peroxiredoxin 1 (PRDX1), Tu translation elongation factor, mitochondrial (TUFM), and cleavage and polyadenylation-specific factor 3 (CPSF3). We drew a forest plot based on these eight prognostic characteristic genes (Fig. 2D). Pearson correlation analysis showed that the prognostic genes were expressed independently of each other, providing further evidence of their potential role as independent risk factors for LUAD (Fig. 2C). To clarify the impact of eight prognostic-related genes on OS of LUAD patients, we identified the expression levels of eight genes in the disease group samples in TCGA-LUAD dataset, divided the eight genes into the high- and low-expression groups according to the median expression, and performed survival analysis for the eight prognostic-related NRGs and plotted a Kaplan–Meier survival curve (Fig. 3). The results indicated that four (CPSF3, OGT, PRDX1, and DDX17) of these eight genes were significantly associated with the OS of patients, further confirming their prognostic value.

Next, to explore the differential expression of eight genes (CALM1, DDX17, FPR1, OGT, PGLYRP1, PRDX1, TUFM, CPSF3) in the model, we stratified the LUAD group into high-risk and low-risk groups based on the median risk core in the Cox model. The differential expression of the eight genes was analyzed, and a scatter plot was drawn to visualize the distribution. In the TCGA-LUAD dataset, the expression levels of CALM1, FPR1, OGT, DDX17, and PGLYRP1 were significantly increased in the high-risk group, whereas that of CPSF3 was significantly increased in the low-risk group. In the GSE68465 dataset, the expression levels of CALM1, FPR1, OGT, PRDX1, DDX17, and PGLYRP1 were significantly increased in the high-risk group, whereas that of CPSF3 was significantly increased in the low-risk group. The results show that the expression trends of CPSF3, CALM1, FPR1, OGT, DDX17, and PGLYRP1 were consistent and significant in the

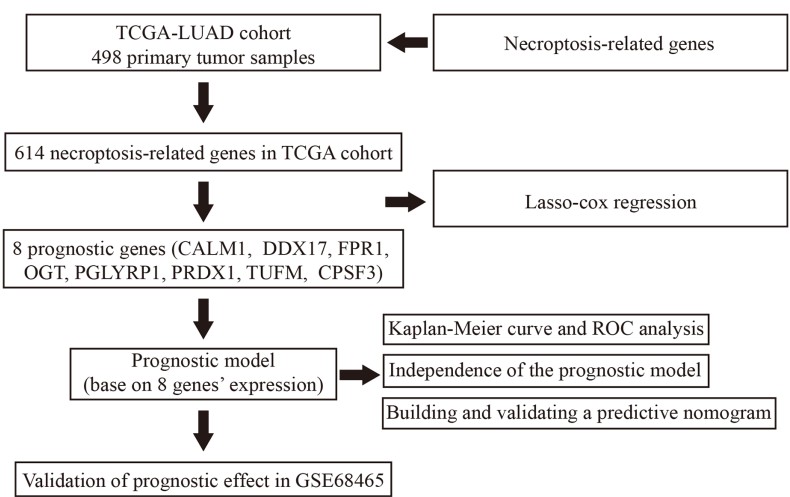

**Figure 1 Flowchart illustrating the overall study design and analysis process.**

TCGA-LUAD and GSE68465 datasets (Fig. 4). In summary, these genes were differentially expressed between the high- and low-risk groups, providing a basis for prognostic evaluation.

## Functional enrichment analysis

To further explore the functions of NRGs, we utilised the GO and KEGG databases to perform functional enrichment analysis of differentially expressed genes in patients with a high or low risk for LUAD. The differentially expressed genes of both groups were mainly enriched for GO biological process (BP) terms such as programmed necrotic cell death, NF-κB signalling pathway, and response to tumour necrosis factor (Fig. 5A). In terms of cellular components (CCs), we observed significant enrichment for several GO terms such as CD40 receptor complex, membrane microdomain, and focal adhesion (Fig. 5B). In addition, several GO terms related to molecular functions (MFs) were significantly enriched, such as ubiquitin protein ligase binding, protein folding chaperone, protease binding, and tumour necrosis factor receptor binding (Fig. 5C).

KEGG-based enrichment analysis revealed several pathways significantly enriched among the differentially expressed genes, including those associated with NOD-like receptor signalling pathway, apoptosis, TNF signalling pathway, NF-κB signalling pathway, necroptosis, lipid and atherosclerosis, influenza A, Kaposi sarcoma-associated herpesvirus infection, C-type lectin receptor signalling pathway, and RIG-I-like receptor signalling pathway (Fig. 5D). The functional enrichment analysis of differentially expressed genes revealed the biological processes and pathways associated with NRGs. These results helped us understand the function of these genes in LUAD.

Next, we performed GSEA to further explore the mechanism of NRGs in LUAD. The results revealed that LUAD is mainly enriched in the extrinsic component of membrane, lymphocyte differentiation, cell morphogenesis, I-κB kinase/NF-κB signalling, and immune system process (Figs. 6A and 6B), as well as KEGG pathway terms such as

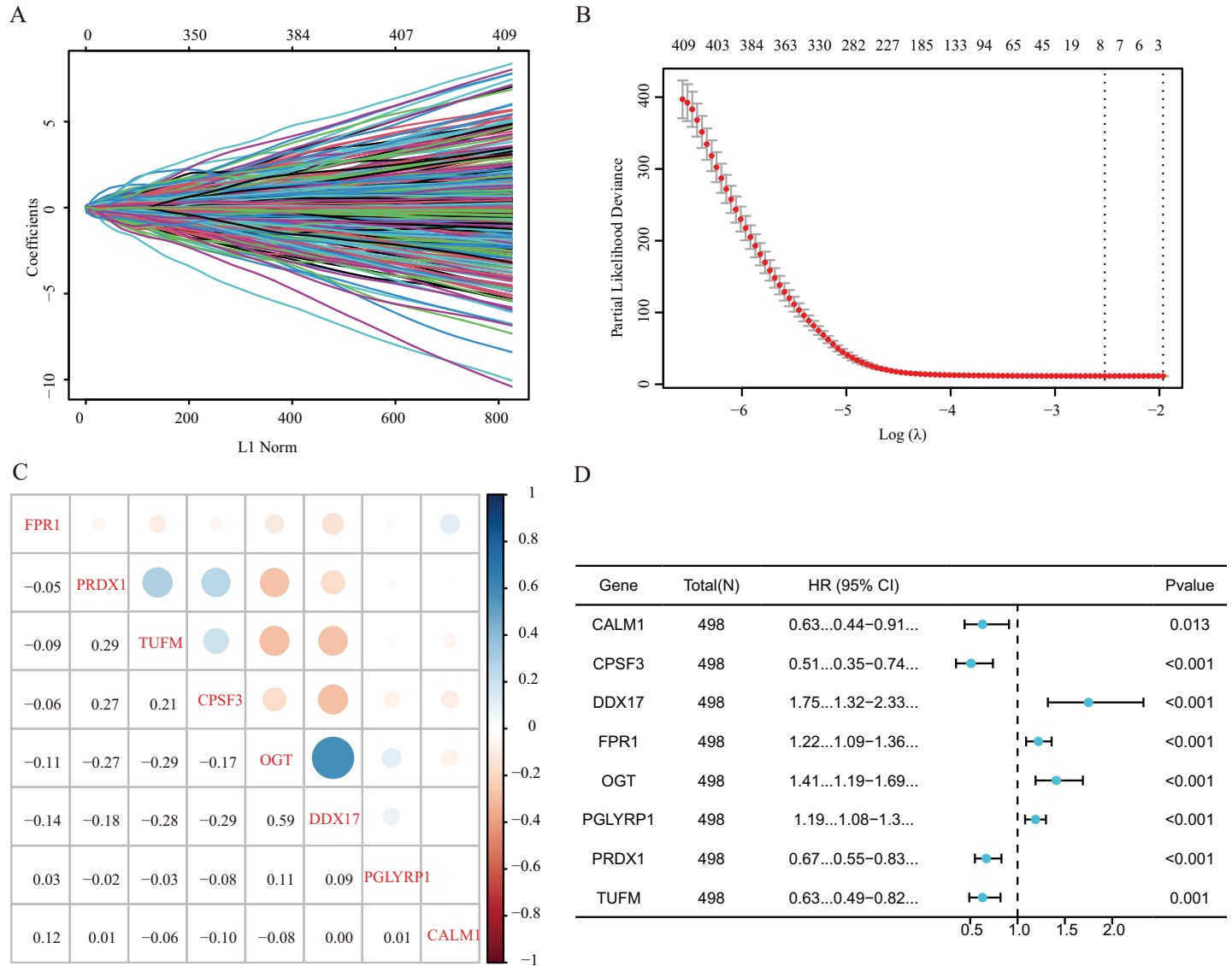

**Figure 2 Establishment of a prognostic gene signature *via* LASSO Cox regression analysis.** (A) LASSO variable locus diagram. The LI Norm represents the size of the regularised penalty parameter. The coefficients of the corresponding variables reflect the importance of each variable in model prediction. With the increase in regularization penalty parameters, the coefficients of some variables may gradually approach zero, facilitating variable selection and reducing the model complexity. In the Lasso-Cox regression analysis, for each gene regression coefficient, positive and negative numbers represent positive and negative correlations, respectively. (B) LASSO coefficient screening to select the best parameter λ. λ represents the size of the regularization penalty parameter, which is usually used to control the sparsity of the model. The red dots represent the best model chosen for these different λ values, whereas grey lines represent model performance at different λ values. (C) Pearson correlation coefficients analysis for the selected gene-expression values, where the colour represents the Pearson correlation coefficient, and the size represents the absolute value of the coefficient. (D) Forest map of eight prognostic characteristic genes.

lysosome, rheumatoid arthritis, arrhythmogenic right ventricular cardiomyopathy, adherens junction, and *Staphylococcus aureus* infection (Figs. 6C and 6D). These GSEA results showed significant enrichment for biological functions related to cell morphology changes, which supported the GO and KEGG data regarding the biological functions and pathways associated with the NRGs. GSEA further revealed the biological processes and

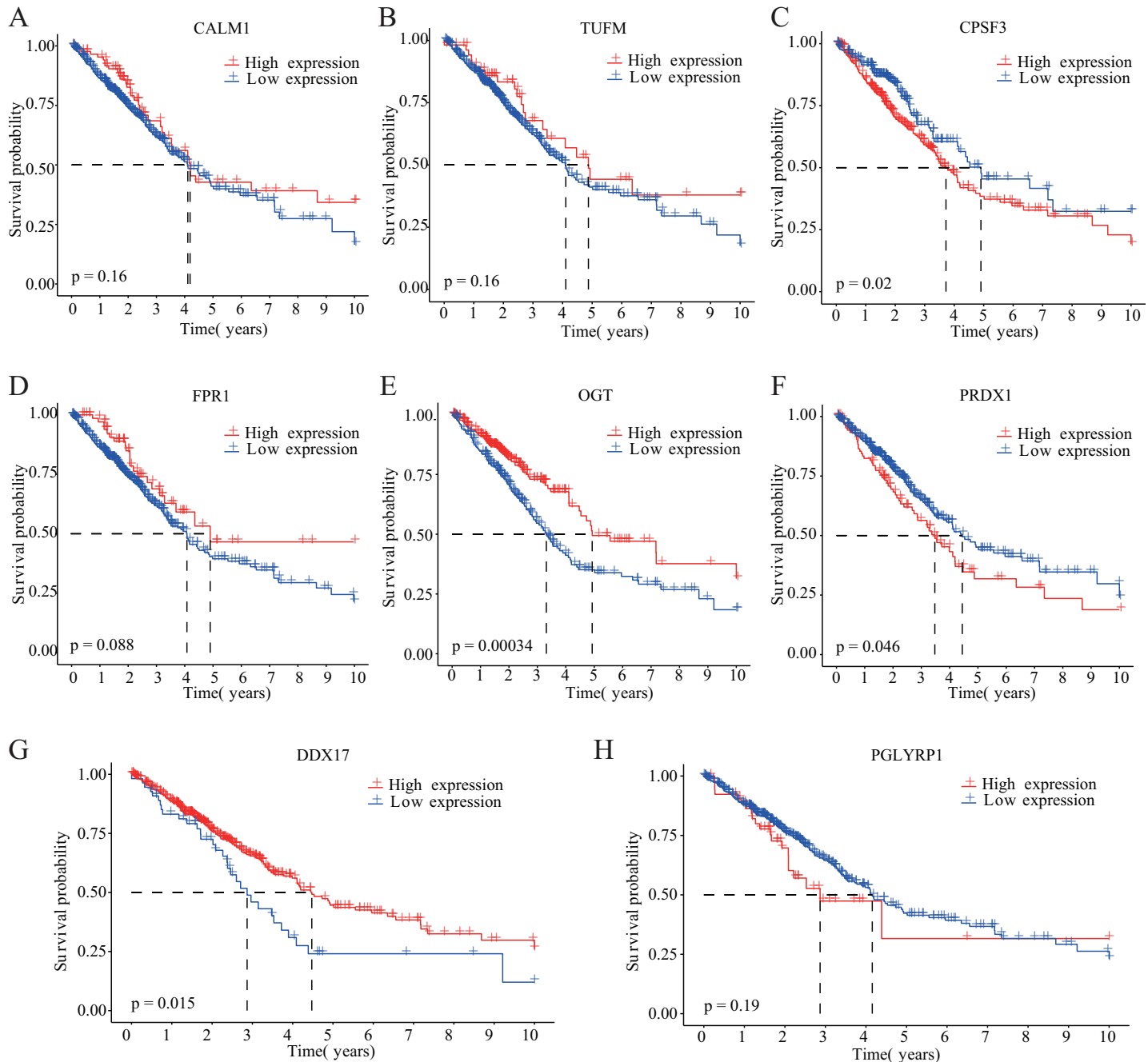

**Figure 3 Kaplan–Meier analysis of the correlations between eight NRGs and LUAD prognosis.** (A–H) Survival curves are shown for each NRG between the high-expression and low-expression groups, including 498 TCGA-LUAD samples with prognostic information.

enriched pathways related to NRGs in different risk groups. This helped us gain a deeper understanding of the roles of these genes.

GSVA was further used to study differentially expressed NRGs in the high- and low-risk groups (Fig. 7A). We identified GO BP terms that were upregulated (*e.g.*, positive regulation of epithelial cell proliferation and regulation of epithelial cell apoptotic process

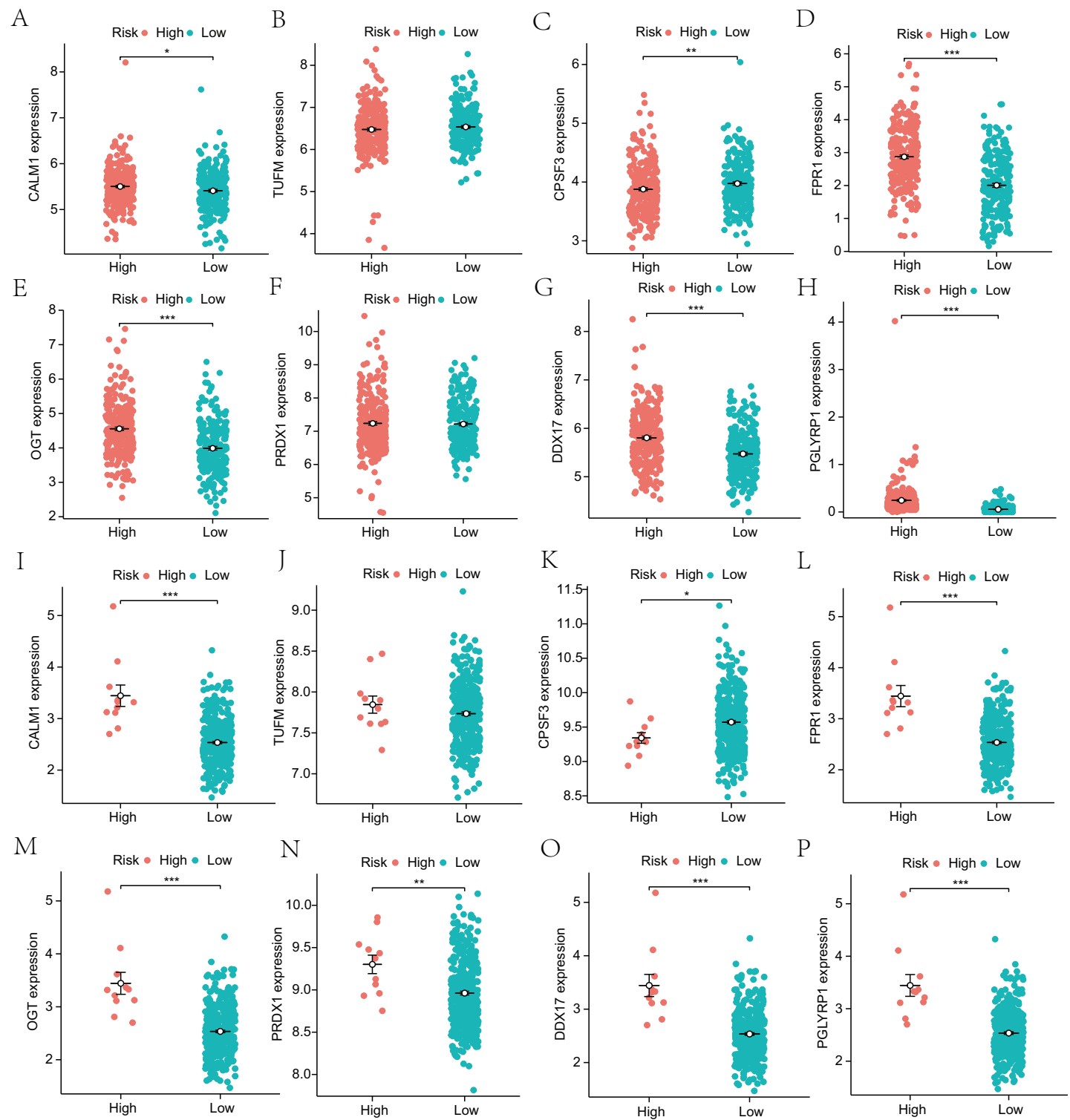

**Figure 4 Analysis of differential expression of the identified prognostic genes between patients in the high-risk and low-risk groups.** (A–H) TCGA-LUAD training dataset. A total of 498 LUAD samples were included. (I–P) GSE68465 validation dataset. A total of 371 LUAD samples were included. $*p < 0.05$, $**p < 0.01$, $***p < 0.001$.

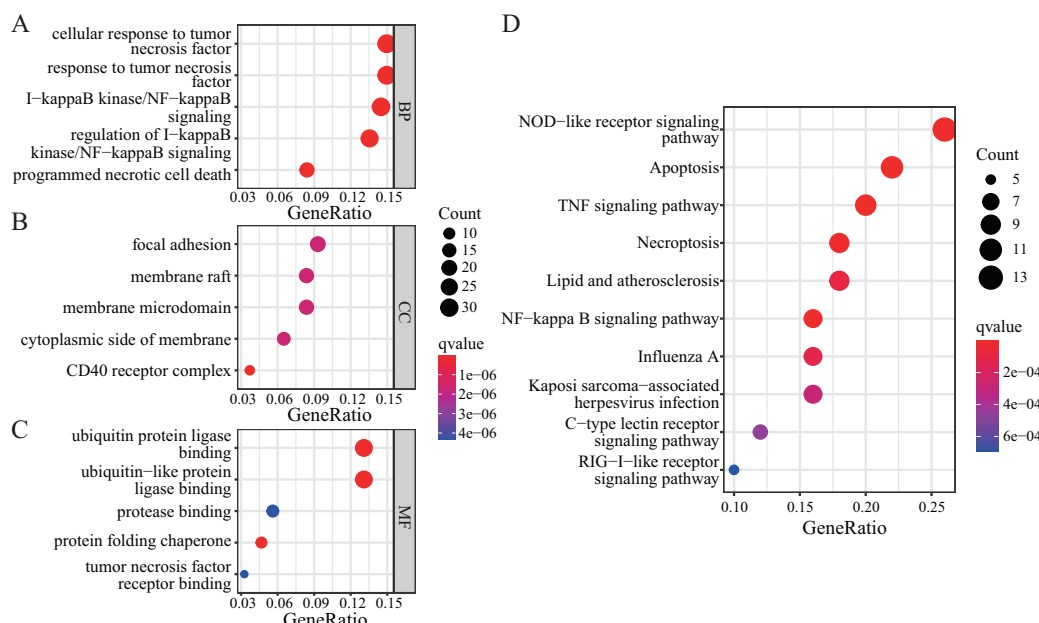

**Figure 5 GO and KEGG enrichment analyses of NRGs.** (A) The top five BPs. (B) The top five CCs. (C) The top five MFs. (D) The top ten KEGG pathways.

function) and downregulated (*e.g.*, regulation of cellular localisation, regulation of DNA metabolic process, and regulation of chromosome organisation) in the high-risk group (Figs. 7B and 7C). GSVA revealed the biological enrichment of these genes in the high- and low-risk groups, further elucidating their role in LUAD.

## NRG network construction

PPI analysis was conducted to further understand the MFs associated with differentially expressed NRGs (Fig. 8A). The PPI network contained 120 NRGs and 732 associated lines, and the mean local clustering coefficient was 0.672. Four key subnetworks were extracted using the MCODE plug-in of Cytoscape software. We selected the top three key subnetworks (consisting of 40 hub genes) for further analysis (Figs. 8B–8D). Through PPI network analysis, we identified the interaction network of NRGs and identified hub genes. This helped us understand the importance of these genes in protein interactions.

## Assessment of the risk-scoring system

All patients were equally divided into a low- or high-risk group based on the median risk score. The survival-time distribution indicated that patients in the high-risk group had a higher mortality rate (Fig. 9A). The expression levels of eight NRGs are also shown in the heat map in Fig. 9A. In addition, the Kaplan–Meier curves illustrated that patients with lower risk scores had a better prognosis than did those with higher scores (Fig. 9B, $p = 0.00013$). The AUCs of the time-dependent ROC were 0.51, 0.60, or 0.61 after survival for 1, 3, or 5 years, respectively, which showed that the predictive power of the model increased over time (Fig. 9C). Multivariate Cox analysis showed that metastasis, gender, and risk scores had prognostic value for patients with LUAD (Fig. 9D). Multivariate Cox
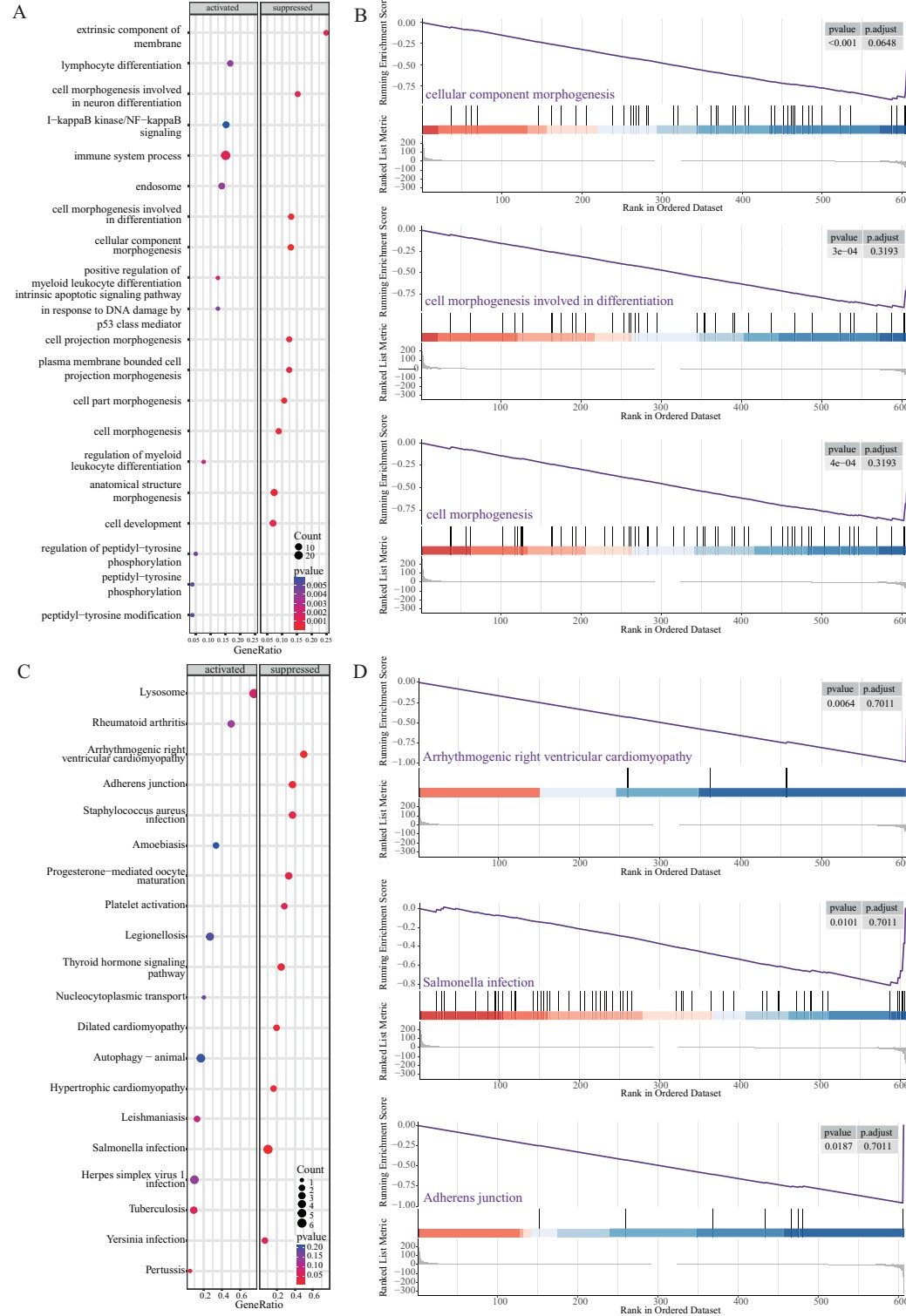

**Figure 6 GSEA of NRGs in the high-risk and low-risk groups.** (A) GSEA-based GO analysis of TCGA cohort. (B) The top three categories identified by GSEA-based GO analysis of TCGA cohort. (C) GSEA-based KEGG analysis of TCGA cohort. (D) The top three pathways identified by GSEA-based KEGG analysis of TCGA cohort.

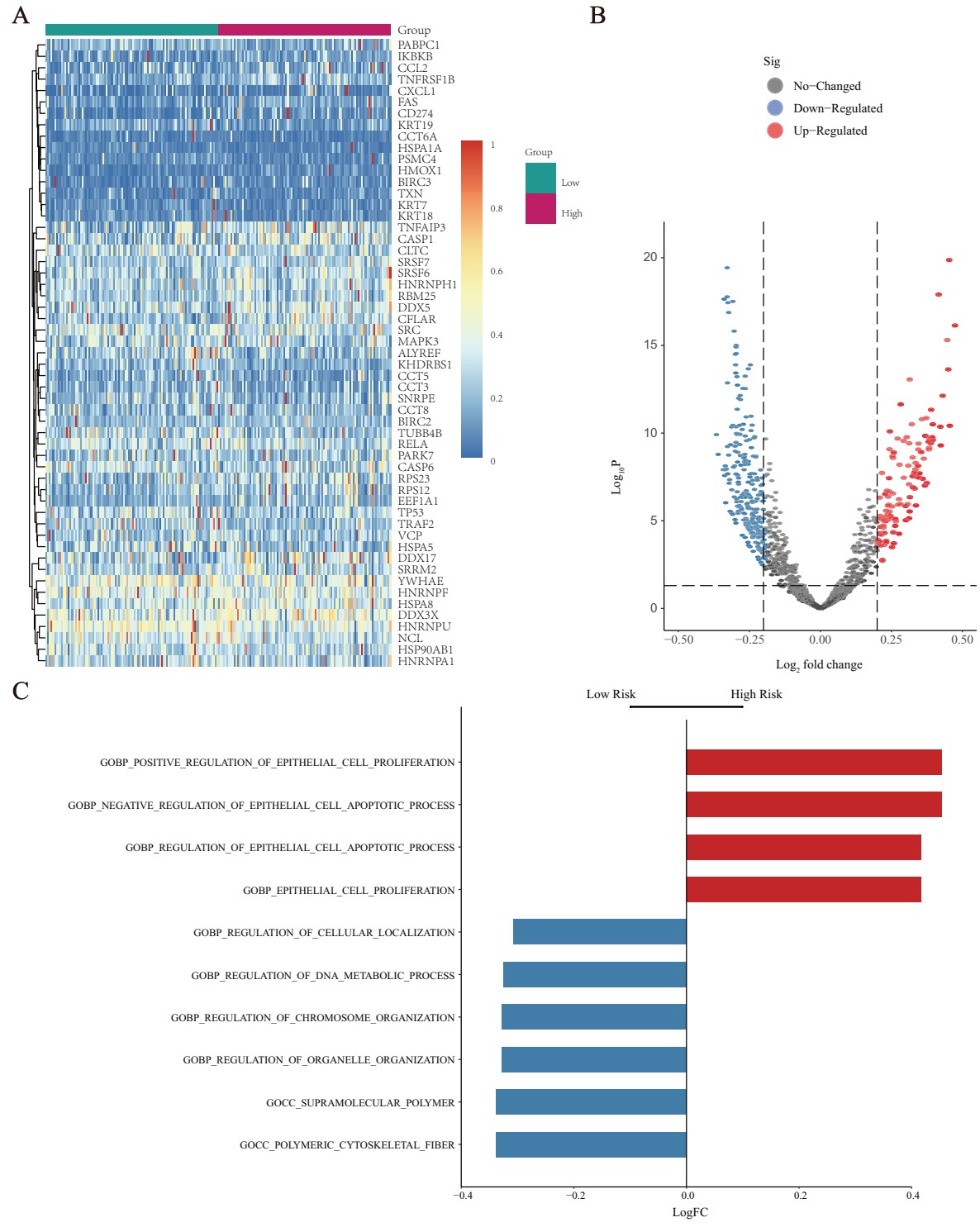

**Figure 7** **GSVA of NRGs in the high- and low-risk groups.** (A) Heatmap of differentially expressed genes (DEGs) identified by GSVA, after normalization for gene-expression levels. (B) Volcano plot of for the genes identified by GSVA, where the vertical axis represents -$\log_{10}$ ($p$ value), whereas the horizontal axis represents the -$\log_2$ fold-change (FC) of enrichment fraction. (C) The histogram for the GSVA analysis indicates variations in pathways, where the ordinate represents different pathways, the abscissa represents the logFC, and the data were processed by regularization. The red colour indicates upregulation in the high-risk group, and the blue colour indicates downregulation in the high-risk group.

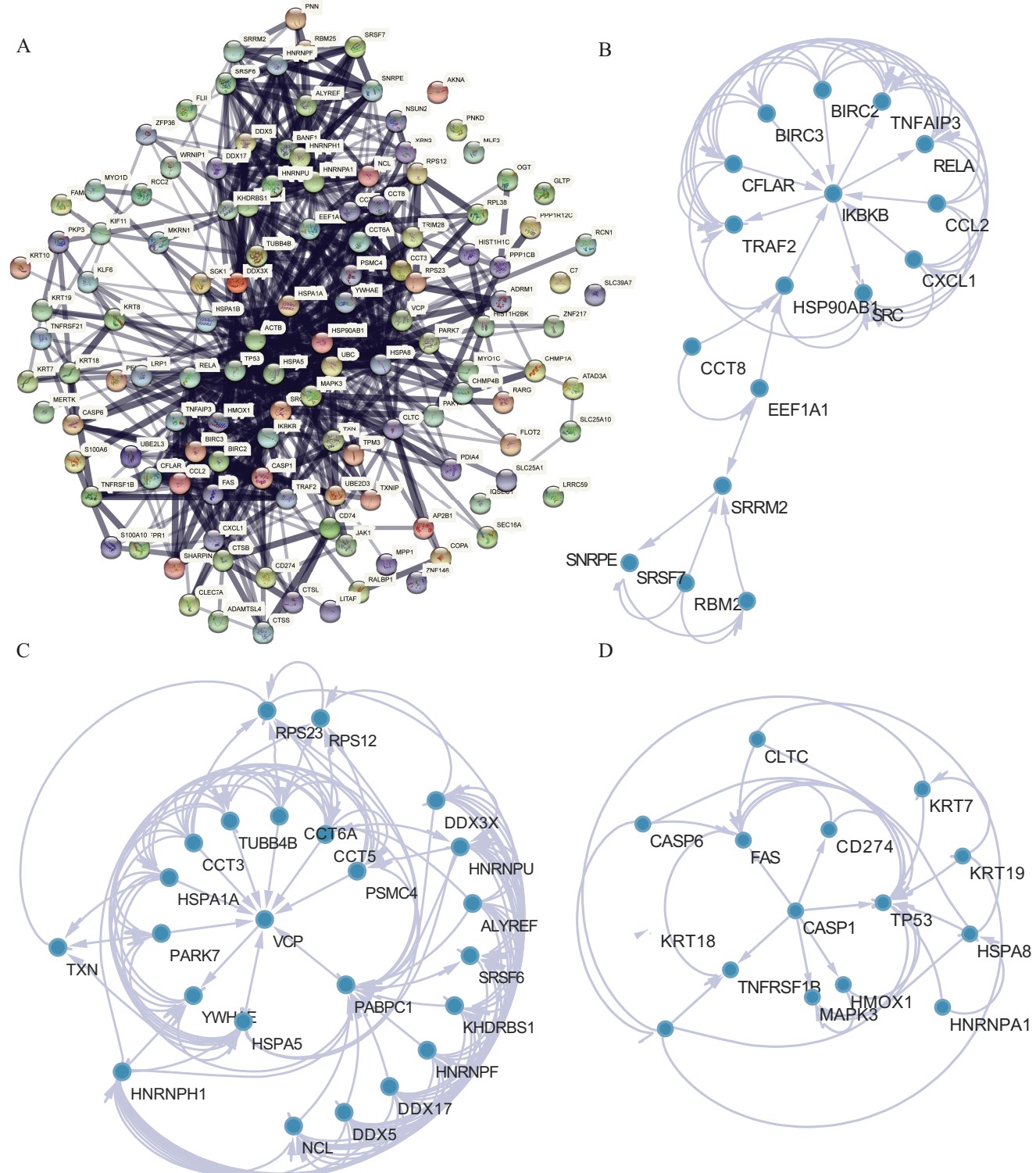

**Figure 8 PPI network analysis.** (A) PPI network. (B–D) Subnetworks 1–3 of the PPI network.

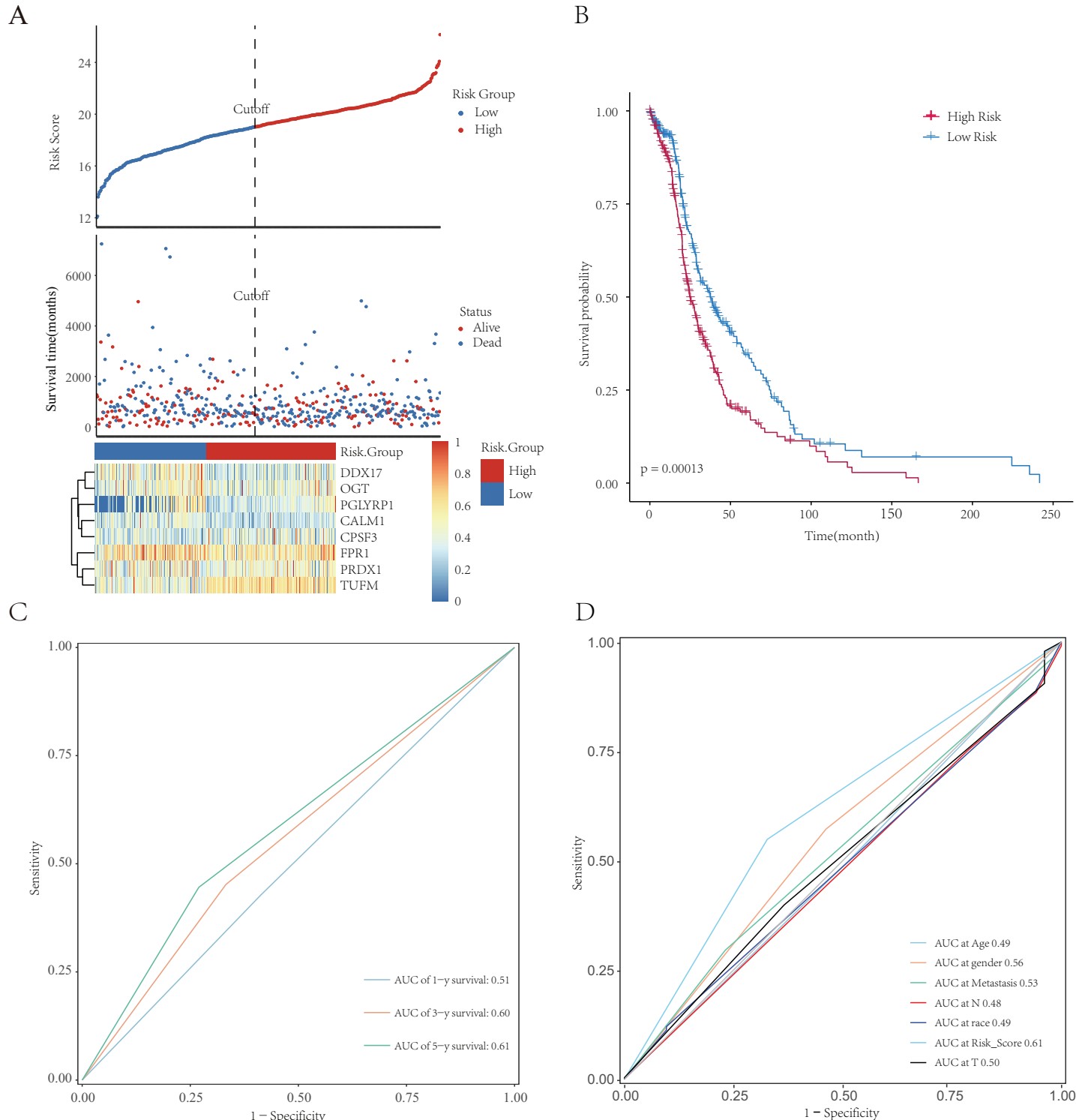

**Figure 9 Establishment and evaluation of prognostic models. The analysis included 498 LUAD samples.** (A) The risk score, survival-time distributions, and gene-expression heat map of immune-related NRGs in the TCGA-HCC cohort. (B) Kaplan–Meier survival curves related to the eight-gene signature between the high- and low-risk groups in the training set. (C) The ROC curve shows the potential of NRGs signature in predicting 1-, 3-, and 5-year OS rates for the training set. (D) The AUC of the ROC was used to compare the prognostic accuracy of the risk score and other prognostic factors for the training set.

regression analysis identified eight independent risk genes and we established a risk-scoring model. Survival analysis showed that the prognosis of patients in the high-risk group was significantly worse than that in the low-risk group.

## Validation of clinical prognostic value of the model

We integrated several clinical factors to develop a stable and reliable clinical prediction model for LUAD. The results of multivariate Cox analysis indicated that metastasis and the risk score were independent risk factors (Fig. 10A). We further explored the performance of the model by conducting survival analysis and showed that both the M0 and M1 stages of LUAD were significantly associated with the OS of LUAD patients ($p < 0.05$; Figs. 10B and 10C). Moreover, we plotted time-dependent ROC curves to evaluate the performance of the clinical predictive model. The AUCs were 0.58, 0.60, or 0.63 after 1, 3, or 5 years of survival, respectively, for patients in the internal cohort (Fig. 10D), whereas those for patients in the GSE68465 validation cohort were 0.61, 0.54, or 0.53, respectively (Fig. 10E). We constructed a clinical prediction model for LUAD based on clinical factors and evaluated its performance through internal and external validation. This model could be used to predict the survival of patients.

## Establishment and evaluation of the nomograms

We established nomograms with independent factors identified through multivariate Cox regression analysis of the clinical prediction model (Fig. 11A). The prediction curves for 1-, 3-, and 5-year OS showed that the accuracy of the nomograms increased over time (Figs. 11B–11D). Therefore, the nomograms further confirmed the reliability and prospective clinical applicability of the risk model.

## Validation and assessment of NRG expression levels in LUAD tissues

To further evaluate the expression levels of NRGs in LUAD and normal lung tissues, we obtained immunohistochemical staining images for each characteristic NRG in the HPA Database. CALM1, CPSF3, FPR1, OGT, DDX17, and PGLYRP1 were expressed at significantly different levels between LUAD and normal lung tissues (Fig. 12). Next, we conducted RT-qPCR experiment to verify the mRNA expression levels of NRGs in cell lines. As shown in Fig. 13, the mRNA expression levels of CALM1, PRDX1 and PGLYRP1 were markedly lower in normal 2B lung cells than in A549 and H1650 LUAD cells. Through immunohistochemical analysis and RT-qPCR experiment, we demonstrated the differential expression of prognostic-related NRGs in LUAD.

## DISCUSSION

The major clinical challenges in treating LUAD are limited treatment options and the frequent diagnosis at a late stage, which often lead to poor outcomes for patients. In cases where patients are diagnosed with LUAD at an advanced stage, the opportunity for surgical resection has usually passed. Currently, targeted therapies are some of the more effective treatments for LUAD. Necroptosis plays an important role in the pathogenesis of various tumours, such as leukaemia (*Wu et al., 2014*), melanoma (*de Almagro et al., 2015*), and breast cancer (*Stoll et al., 2017*). Recent findings showed that targeting necroptosis

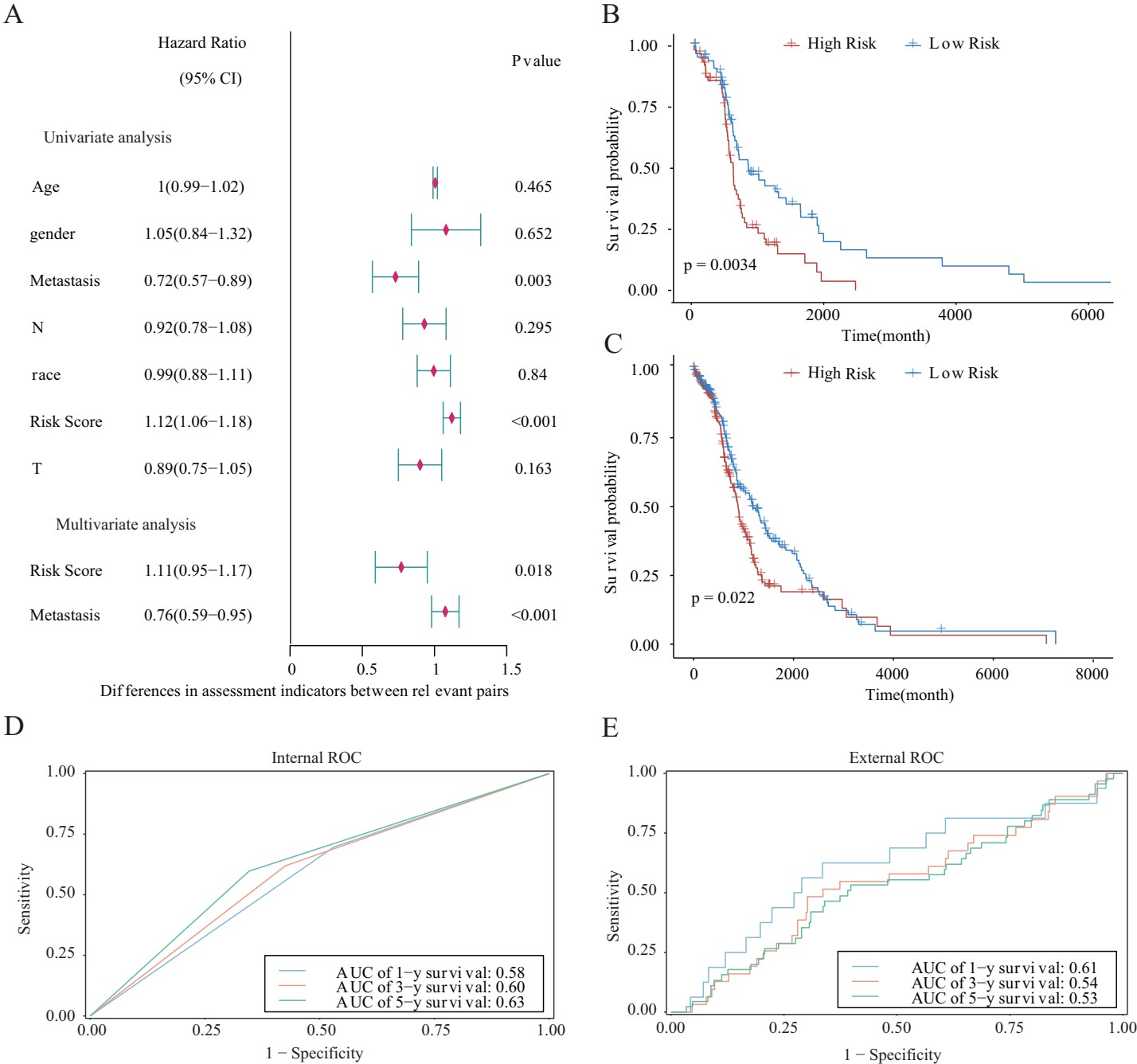

**Figure 10 Prognostic risk score as an independent predictor of overall survival for patients with LUAD.** (A) Forest map showing the results of univariate and independent multivariate Cox regression analysis for the prognostic model of the entire training cohort. (B, C) Kaplan–Meier survival curves for patients with (B) M0-stage LUAD or (C) M1-stage LUAD. (D) Potential for internal validation of 1-, 3-, and 5-year OS in the TCGA-LUAD datasets. A total of 498 LUAD samples were included for analysis. (E) Potential for external validation of 1-, 3-, and 5-year OS in the GSE68465 dataset. A total of 371 LUAD samples were included for analysis.

with various drugs that can manipulate the necroptotic pathway may be a novel and viable approach for anti-tumour therapy (*Xu et al., 2017*). In addition, the expression levels of NRGs can affect the OS of patients with cancer.

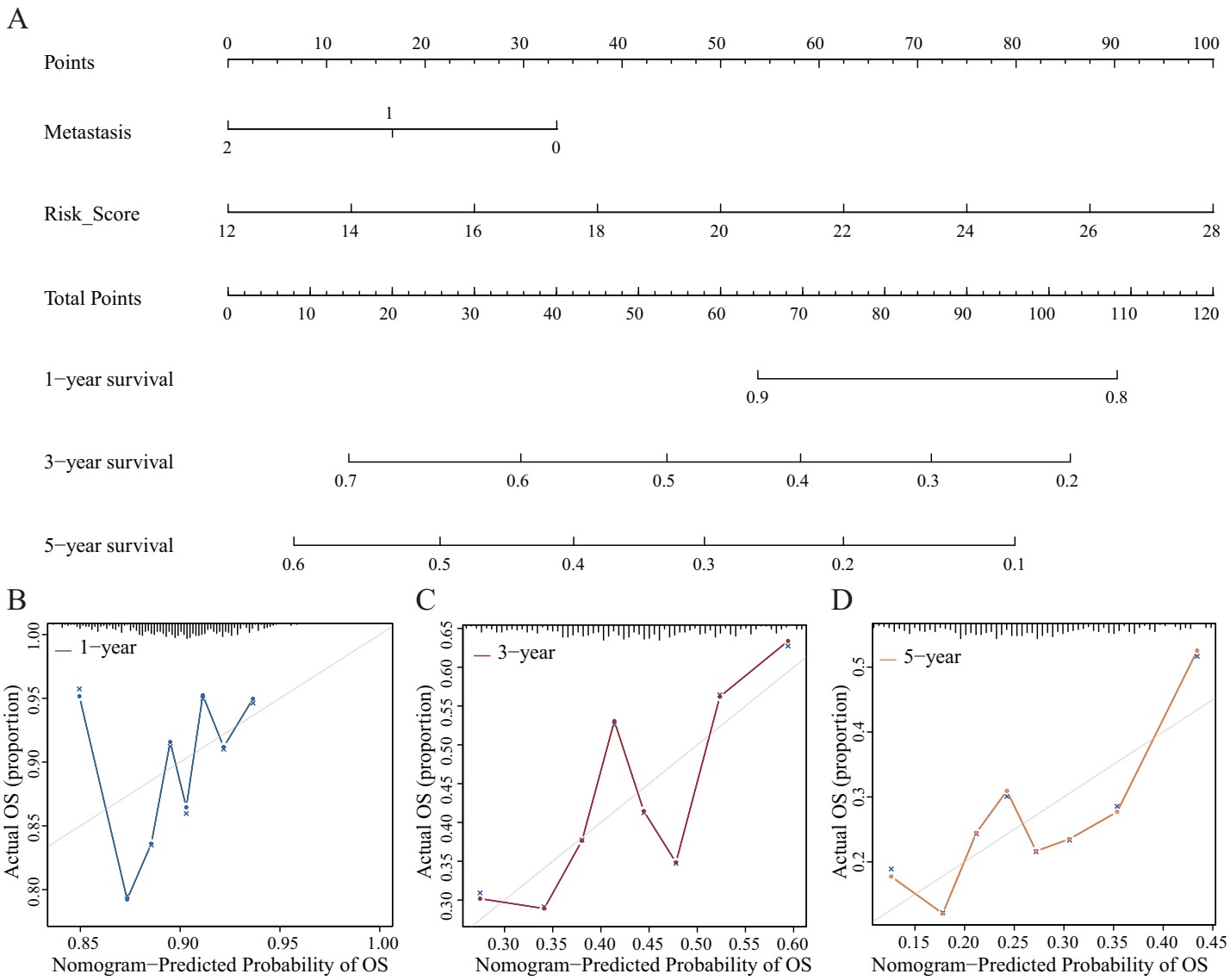

**Figure 11 Nomograms and decision curve analysis (DCA) for the prognostic model.** (A) The nomograms were used to predict the 1-, 3-, and 5-year OS rate of patients with LUAD. (B–D) DCA was used to predict the OS rate of patients with LUAD.

This study mainly focused on elucidating the potential mechanism of necroptosis and establishing a corresponding prognostic model for LUAD. Previous studies have mainly focused on single-analysis methods (*Song et al., 2022*). By contrast, we integrated transcriptome data of large-scale LUAD samples from TCGA and employed various analysis methods including GSVA, GO, KEGG, survival analysis, and experimental validation to explore the role of NRGs in LUAD from multiple perspectives. Our study could eventually help physicians to deploy new treatment targets and assess prognosis.

We identified NRGs and obtained mRNA expression data (and the associated clinical information) for LUAD samples from the GeneCards database and TCGA, respectively, and used LASSO Cox regression and multivariate Cox analysis to identify eight hub NRGs,

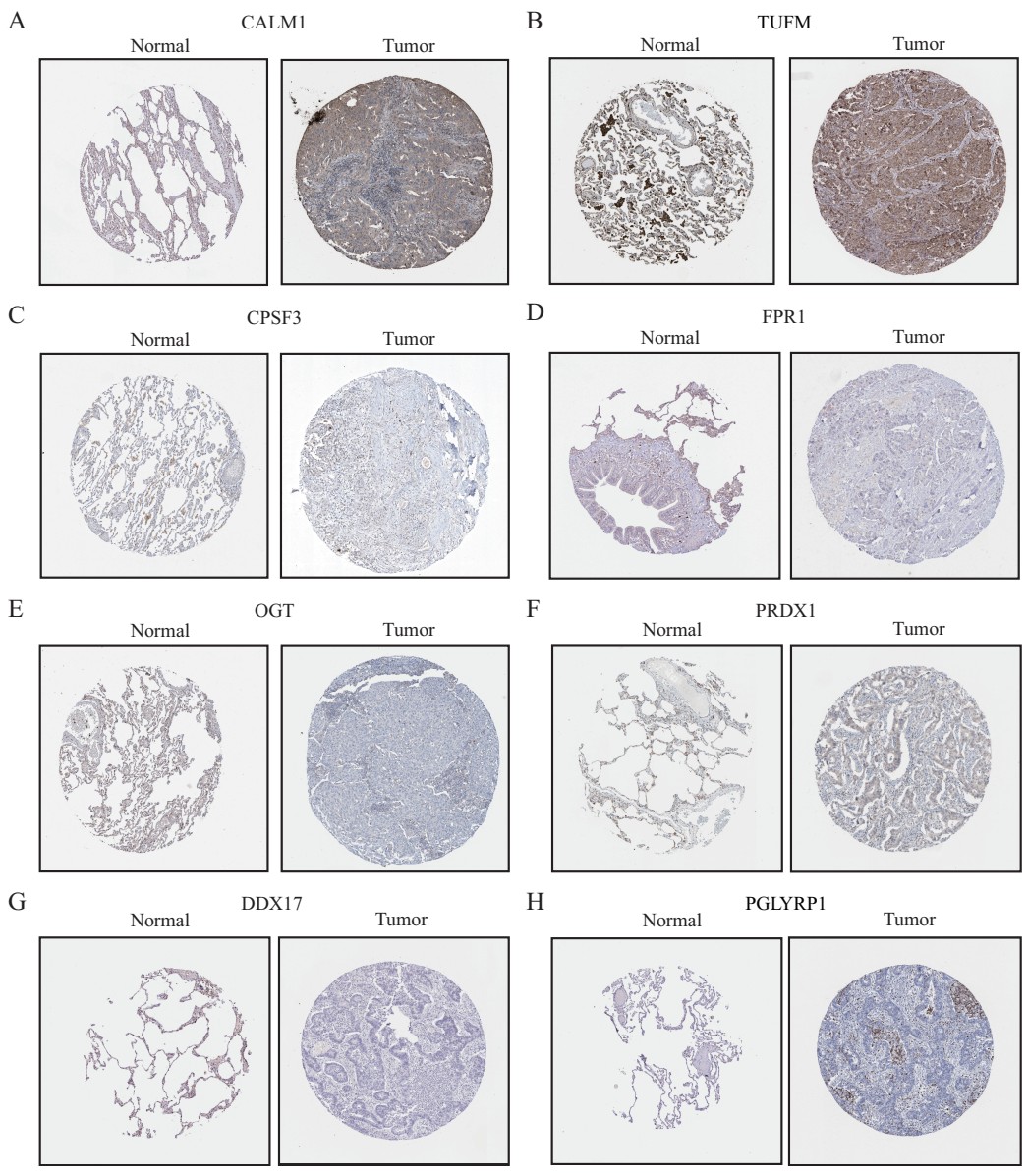

**Figure 12 The protein-expression levels of eight NRGs in the HPA database based on immunohistochemistry analysis.**

which were used to construct a prognostic risk model. The model showed accurate predictions with an internal test cohort and an external validation cohort. Moreover, a nomogram was established to predict the probabilities of 1-, 3-, and 5-year OS. Furthermore, the NRGs showed significant expression differences between LUAD cells and normal lung cells in RT-qPCR validation experiments.

Eight NRGs (CALM1, DDX17, FPR1, OGT, PGLYRP1, PRDX1, TUFM, and CPSF3) that were identified as independent prognostic indicators for patients with LUAD were selected for the prognostic prediction model. CALM1 acts in the calcium signal-transduction pathway (*Kobayashi et al., 2015*) that was differentially expressed

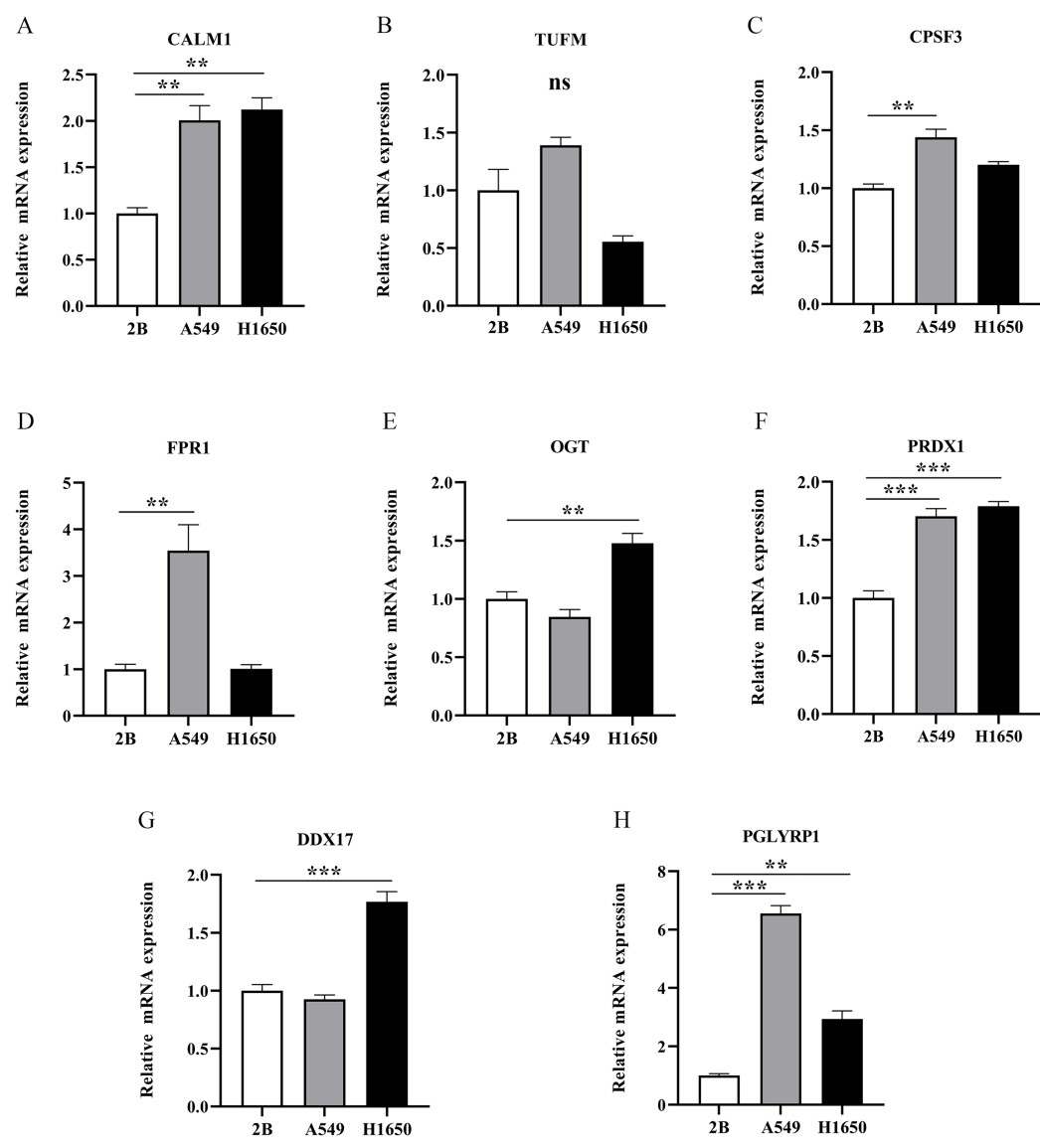

**Figure 13 The mRNA-expression levels of eight NRGs in LUAD cells and normal lung cells.** $**p < 0.01$, $***p < 0.001$.

owing to DNA methylation in multiple cancers. CALM1 downregulation has high diagnostic and prognostic potentials for lung cancer (*Yao et al., 2021*).

DDX17 participates in multiple cellular processes as an RNA helicase, including alternative splicing and microRNA processing (*Terrone et al., 2022*). Previous data showed that DDX17 could regulate alternative splicing and produce oncogenic molecules that promote hepatocellular carcinoma metastasis, indicating that DDX17 expression was strongly associated with patient outcomes (*Zhou et al., 2022*).

FPR1 is a chemotactic factor produced by neutrophils involved in innate and adaptive immunity (*Kuley et al., 2021*). The pharmacologic inhibition of FPR1 decreased T cell migration and infiltration into tumour microenvironments in most patients with locally

advanced rectal cancer harbouring the CC genotype (E346A) of FPR1 (*Chiang et al., 2021*), highlighting it as an independent predictor.

Recent data implicated OGT as a key molecule for tumour metastasis and chemoresistance (*Shi et al., 2022*), such as with breast (*Liu et al., 2022*) and ovarian (*Zhou et al., 2018*) cancer. OGT is a potential therapeutic target for some cancers, including small-cell lung cancer (*Tang et al., 2022*). A previous study showed that an OGT inhibitor significantly inhibited breast cancer cell invasion and metastasis (*Liu et al., 2022*).

The PGLYRP1 protein regulates innate immunity and plays a central role in antitumor-defence systems. Previous findings showed that PGLYRP1 could interact with Hsp70 to form a stable complex that was cytotoxic against some tumour cell lines and promoted apoptosis and necroptosis (*Yashin et al., 2015*). *Walraven et al. (2021)* showed that PGLYRP1 expression was a potential biomarker of the platelet proteome markedly upregulated after anticancer treatment.

Aberrant PRDX1 expression occurs in numerous cancers, particularly in breast, oesophageal, and lung cancers (*Ding, Fan & Wu, 2017*). PRDX1 can promote the epithelial–mesenchymal transition in head and neck squamous cell carcinoma after entering the nucleus (*Jiang et al., 2019*). PRDX1 promotes cell proliferation by activating Wnt–β-catenin signalling and is an independent prognostic factor for disease recurrence and reduced survival in patients with non-small-cell lung carcinoma (*Song et al., 2023*).

TUFM downregulation can induce metastasis and proliferation of lung cancer cells *via* the AMP-activated protein kinase signalling pathway (*He et al., 2016*; *Ashrafizadeh et al., 2021*). TUFM collaborates with ubiquitin-specific peptidase five to regulate the growth of colorectal cancer cells and can serve as a new prognostic indicator for colorectal carcinoma (*Shi et al., 2012*; *Xu et al., 2019*).

CPSF3 plays an important role in inducing cell death (*Zhu et al., 2009*) and is associated with patient prognosis and cancer recurrence in multiple cancers, including bladder cancer (*Xiong et al., 2022*) and LUAD (*Ning et al., 2019*). These findings suggest that the NRGs identified in this study play crucial roles in cancer necroptosis and are relevant to the prognosis of patients with LUAD.

We divided patients with LUAD into low-risk and high-risk groups, based on the median risk score. Patients in the low-risk group showed better outcomes. The predictive performance of our model increased over time, as shown in time-dependent ROC verification experiments. After incorporating the patient's clinical information into our model, multivariate Cox analysis showed that both metastasis and the risk score were independent risk factors. The clinical prognosis model accurately predicted 1-, 3-, and 5-year survival rates of patients in the internal cohort (TCGA), as confirmed using an external validation cohort (GSE68465). Subsequent quantitative analysis of nomograms yielded consistent results. Our screening procedure differed from that of a previous study (*Song et al., 2022*); thus, we identified a different NRG signature and drew different conclusions. Although reports on NRGs in LUAD have been documented, the research focuses of these studies vary. *Zhao et al. (2022)* and *Liu et al. (2022)* focused on the impact of necrosis on the tumour microenvironment of LUAD, while *Lu et al. (2022)* focused on the impact of necrotic transcriptome lncRNA on the prognosis of LUAD. Our research

focused on using necrotic genes to construct prognostic and clinical predictive models for LAUD. The study provides deeper insights that can help clinicians better predict patient survival and guide treatment decisions. In summary, the proposed NRG signature suggests novel targets for cancer treatment strategies, which merit further study.

To gain insight into the potential pathogenic roles of NRGs in LUAD, we mainly focused on strong correlations between NRGs and the immune system or the NF-κB signalling pathway, using GSEA, GO and KEGG analyses, and GSVA. Previous data indicated that NF-κB signalling played a key role in coordinating the expression of genes related to immune responses, consistent with our research results. Since NF-κB and related factors can activate and regulate key molecules related to inflammation in cancer, substantial research is ongoing to assess the potential of NF-κB and related proteins as potential therapeutic targets in some cancers (*Li & Verma, 2002*). The results of a recent study showed that some NF-κB antagonists might have good prospects for inhibiting lung cancer (*Rasmi, Sakthivel & Guruvayoorappan, 2020*).

Finally, we verified the mRNA expression levels of all eight NRGs *via* RT-qPCR. The mRNA expression levels of CALM1, PGLYRP1, and PRDX1 were higher in LUAD cells than in normal lung cells. Our results confirmed that elevated PRDX1 expression in patients with LUAD was associated with a low survival rate. In addition, previous data revealed PRDX1 upregulation as an independent prognostic factor for disease recurrence and a therapeutic target in lung cancer (*Kim et al., 2007*, *2008*). This study represents the first demonstration that CALM1 and PGLYRP1 were significantly overexpressed in LUAD and that they might serve as important genes in future prognostic models. In addition, CALM1 and PGLYRP1 might play crucial roles in the occurrence and development of LUAD. Previous studies (*Zhang et al., 2022*; *Wu et al., 2022*) solely relied on bioinformatics analysis. However, we performed RT-qPCR experimental validation, which confirmed our findings from bioinformatics analysis, thus increasing the credibility of our results.

This study has some limitations. Our research was mainly based on information deposited in two public databases; therefore, our findings require further validation in prospective studies or multi-centre external evidence, including single-cell sequencing. Although bioinformatics analysis of the biological characteristics and mechanism of NRGs in regulating LUAD can provide new avenues for future research, such findings need to be verified by conducting additional *in vitro* and *in vivo* experiments. In the future, we intend to conduct further experiments to determine key LUAD biomarkers and their mechanisms of action.

## CONCLUSIONS

The NRG signature identified in this study provides new ideas for in-depth research and long-term applications in LUAD treatment. CALM1, DDX17, FPR1, OGT, PGLYRP1, PRDX1, TUFM, and CPSF3 are potential prognostic indicators for LUAD. However, further investigation is needed to ascertain the effects of CALM1 and PGLYRP1 on LUAD.

### Funding

This work was supported by the Chengdu Commission of Health Foundation (No. 2022235). The funders had no role in study design, data collection and analysis, decision to publish, or preparation of the manuscript.

### Grant Disclosures

The following grant information was disclosed by the authors:
Chengdu Commission of Health Foundation: 2022235.

### Competing Interests

The authors declare that they have no competing interests.

### Author Contributions

- Xiaoping Zhou performed the experiments, analyzed the data, prepared figures and/or tables, authored or reviewed drafts of the article, and approved the final draft.
- Ming Zhao performed the experiments, analyzed the data, prepared figures and/or tables, and approved the final draft.
- Yingzi Fan conceived and designed the experiments, authored or reviewed drafts of the article, and approved the final draft.
- Ying Xu conceived and designed the experiments, authored or reviewed drafts of the article, and approved the final draft.

### Data Availability

The original code and gene expression data of lung adenocarcinoma are available in the Supplemental Files.

### Supplemental Information

Supplemental information for this article can be found online at http://dx.doi.org/10.7717/peerj.16616#supplemental-information.

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
