# Peer review of "Identification of a necroptosis-related gene signature for making clinical predictions of the survival of patients with lung adenocarcinoma"

_PeerJ, doi:10.7717/peerj.16616_

## Round 0.1 · original submission · Major Revisions

The reviewers acknowledge the importance of this paper. However, many problems have been pointed out, such as unclear methods and figure legends. It is also pointed out that many similar papers have already been published. For acceptance of this paper, it is essential to clarify the advantages of this paper when compared with previous publications, in addition to improving the description of methods and figure legends.

Reviewer 1 ·

Basic reporting

This article is about identification of necroptosis - related gene signature for making clinical predictions about lung adenocarcinoma and extends a good amount of information around the objective. I appreciate the detailing of the work and writing. The fig 1 helped getting the whole workaround easily,

The wealth of information is rich in the manuscript and delivers a good information of statistical models used.

Experimental design

Clearly mentioned , well executed.

Validity of the findings

The results seemed to be well assessed based on bioinformatics analysis. I have few suggestions though regarding some points where manuscript could have been better for example there is no gene described in heatmap in fig 7 A and 9A. It is not helpful to just put heatmaps without gene names.

Fig 7B has legends riding over genes. Please correct that and also label top DE genes.
Line 106- Kindly mention the version of limma that was used.

Line 153-154 the authors talk about the distribution of the data but do not provide any plots to show the nature of the data to follow or not follow normal distribution.

Additional comments

Kindly go through the citations again. I think there are more articles that could be cited in discussion part.

Reviewer 2 ·

Basic reporting

NO

Experimental design

NO

Validity of the findings

NO

Additional comments

1.The findings from TCGA seq analysis were validated using qPCR, which is a common method for gene expression validation. However, additional validation using other techniques, such as immunohistochemistry or functional assays, would further strengthen the findings.
2.Please add data from single cell sequencing (public databases are also available) to confirm that genes play a role in tumor cells.
3. Generally speaking, the higher the AUC value of the ROC curve is, the better the prediction accuracy is. However, in the training and testing set of the prediction model, the AUC value at1, 3 and 5 years is only 0.6.
4.The model building method in this paper is very common and not novel. Why not use principal component analysis to build the model.
5.Please compare the gene model with existing signatures to show its performance better.
6.The nomogram should be tested by proportional hazard assumption.

Reviewer 3 ·

Basic reporting

Please see "Additional comments".

Experimental design

Please see "Additional comments".

Validity of the findings

Please see "Additional comments".

Additional comments

The authors conducted a series of bioinformatics analyses and molecular experiments to explore how eight necroptosis-related genes could be used to predict the prognosis of lung adenocarcinoma patients. The current study would be much improved if the authors address the following concerns:

------[Major Concerns about FIGURES, METHODS, RESULTS, and/or CONCLUSIONS]
1. Necroptosis-related gene signatures in lung adenocarcinoma appear to be investigated by some published papers (PMID: 36027854, PMID: 35368663, PMID: 36506314, PMID: 36452156, PMID: 35911707, PMID: 35754848, PMID: 36504901, PMID: 35965497, etc.). It would be necessary to pinpoint the research/knowledge gap, which the published papers did not fill but the current study is addressing. In other words, please summarize the novelty of this study, compared with those similar publications.

2. In all FIGURES, it would be clear and more readable to BOTH provide figures with high resolution AND expand on figure legends by explaining the meanings of colors, groups, lines, and abbreviations. These revisions would greatly help readers to understand the results and their implications easily and efficiently. For example,
2.1 In FIGURE 1, it seems confusing that the quantities of primary tumor samples (523) and genes (56672) were not consistent with either RESULTS ("Our study design is represented by the flowchart depicted in Fig. 1. We analysed RNA-seq data from 498 primary LUAD samples in TCGA to identify prognostic genes") or METHODS ("RNA-sequencing expression data for 59 normal and 535 LUAD samples were downloaded from TCGA (https://portal.gdc.cancer.gov/) and analysed. After excluding samples with incomplete clinical information, RNA-seq data for 498 samples with complete clinical information were acquired from TCGA database and used as the training data set"). Please modify FIGURE 1 accordingly.
2.2 In the legend of FIGURE 2A, it would be more informative to mention the meaning of "L1 Norm", lines in different colors.
2.3 In the legend of FIGURE 2B, it would be more informative to briefly introduce what the red dots and grey lines stand for.
2.4 In the bar graphs of FIGUREs 4 and 13, it would be more informative to display individual data points; in other words, please replace bar graphs by EITHER scatter plots with bars OR scatter plots (a pattern like PMID: 34537192, PMID: 37046252, and PMID: 37452367). Bar graphs have been shown to be misleading, because they cannot reveal variation/dispersion within data; instead, scatter plots with bars could be acceptable and scatter plots would be preferable (as confirmed by PMID: 25901488 and PMID: 28974579).
2.5 In the legends of FIGUREs 3, 4, 9, 10, and 13, it would be more rigorous to mention BOTH the sample size (the number of data points) AND whether the data points (in FIGURE 13) were technical or biological replicates.

3. In ABSTRACT:
3.1 In Methods ("Functional analyses were conducted to explore the underlying molecular mechanisms"), it would be clearer and more informative to expand on what the "functional analyses" were.
3.2 In Methods ("Finally, mRNA-expression levels in LUAD cell lines were assessed using reverse transcription quantitative polymerase chain reaction (RT-qPCR) analysis to identify NRGs"), it seems better to change this sentence into "Finally, the mRNA-expression levels of the prognostic signatures in LUAD cell lines were assessed using reverse transcription quantitative polymerase chain reaction (RT-qPCR) analysis". After this revision, the sentence would be clearer and more cohesive (that is, sentences are closely connected).
3.3 In Results ("Areas under the ROC curves and the nomogram were used to predict the prognosis of patients with LUAD, and the curves revealed good predictive abilities"), it would be more informative to rewrite this sentence by quantifying how "good" the predictive abilities were.

4. In RESULTS:
4.1 It would be clearer to end each paragraph in RESULTS with one sentence: "Together, these results suggest that ..." (a pattern like PMID: 37452367, PMID: 34715879, PMID: 34384362, PMID: 35965679, and PMID: 34537192), summarizing a paragraph AND highlighting the implications of all results in the paragraph.
4.2 In "Construction of the necroptosis prognosis signature" ("We built a proportional risk model based on the NRGs (Fig. 2A–B). Univariate and multivariate Cox regression analyses were performed to screen for independent risk factors of necroptosis in LUAD, which identified Calmodulin 1 (CALM1) ..."), it seems not clear how the authors picked the eight genes out from all other genes. In particular, this picking process did not seem to be explicitly revealed in FIGURE 2. Ideally, the presentation of this process would be as intuitive as a heatmap — revealing both how good the select genes are and how bad the other genes are.
4.3 In "Construction of the necroptosis prognosis signature" ("Survival analysis was performed using the Kaplan–Meier Plotter Database, which revealed four genes were associated with OS (Fig. 3)"), it would be more informative and rigorous to rewrite this section by expanding on how the authors set up "the high-expression and low-expression groups", an expression that appeared in the legend of FIGURE 3 but was not explained. The description "All patients in the TCGA-LUAD cohort were equally divided into two groups, namely, those with low and high risk scores (based on the median risk score)" could be not clear enough.
4.4 In "Construction of the necroptosis prognosis signature" ("All patients in the TCGA-LUAD cohort were equally divided into two groups ... whereas CPSF3 and OGT expression were expressed at lower levels in the high-risk group (Fig. 4) ... whereas CPSF3 and OGT were downregulated)"), it would be more informative and clearer to point out the implications of these data — what did these results suggest.

------[Minor Concerns about writing]
1. Throughout the manuscript, it seems better to use Grammarly (https://www.grammarly.com/) to check & correct potential grammatical errors or typos. For example,
1.1 In INTRODUCTION's Paragraph 1 ("Lung adenocarcinoma (LUAD) is the major pathological subtype of lung cancer, approximately accounts for 40% of all lung cancer cases"), it seems better to change this sentence into "Lung adenocarcinoma (LUAD) is the major pathological subtype of lung cancer, accounting for approximately 40% of all lung cancer cases".
1.2 In INTRODUCTION's Paragraph 2 ("Necroptosis, a caspase-independent mode of programmed cell death that is mainly regulated by the core RIPK3 and MLKL proteins (Cai & Liu, 2014), is characterised by cell-membrane rupture; cytoplasmic adenosine triphosphate degradation; the release of damage related molecular modules, cytokines and chemokines"), it seems better to change this sentence into "Necroptosis, a caspase-independent mode of programmed cell death that is mainly regulated by the core RIPK3 and MLKL proteins (Cai & Liu, 2014), is characterised by cell-membrane rupture, cytoplasmic adenosine triphosphate degradation, and the release of damage-related molecular modules, cytokines, and chemokines".
1.3 In INTRODUCTION's Paragraph 3 ("We successfully constructed both risk and prediction models, validated using data for a training cohort (obtained from The Cancer Genome Atlas [TCGA] and an external Gene Expression Omnibus (GEO) validation cohort"), it seems better to change this sentence into "We successfully constructed both risk and prediction models using a training cohort, which was obtained from The Cancer Genome Atlas (TCGA). Furthermore, the models were supported by an external validation cohort from Gene Expression Omnibus (GEO)". After this revision, the sentences would be easier to read.
1.4 In INTRODUCTION's Paragraph 3 ("In addition, we studied the expression of select genes using reverse transcription quantitative polymerase chain reaction (RT-qPCR) analysis, and, using complete clinical information for patients, established a nomogram to calculate the survival probabilities of patients"), it seems better to change this sentence into "In addition, we studied the expression of select genes using reverse transcription quantitative polymerase chain reaction (RT-qPCR) analysis and utilized clinical information about patients to establish a nomogram calculating survival probabilities of patients."

2. In ABSTRACT:
2.1 In Results ("Areas under the ROC curves and the nomogram were used to predict the prognosis of patients with LUAD"), it seems better to change this sentence into "Both areas under the ROC curves and a nomogram were used to predict the prognosis of patients with LUAD". After this revision, the sentence would be clearer (easier to understand).
2.2 In Results ("The RT-qPCR results demonstrated that eight genes, especially CALM1, PRDX1 and PGLYRP1, were differentially expressed in LUAD cells"), it seems better to change this sentence into "RT-qPCR results ...".

---

## Round 0.2 · Minor Revisions

Comments from reviewer 3 are important for improving the manuscript. Please note that it is required for acceptance.

Reviewer 1 ·

Basic reporting

See comments

Experimental design

See comments

Validity of the findings

See comments

Additional comments

Reasonable answers to points raised in first review so I think the manuscript can be accepted

Reviewer 3 ·

Basic reporting

Thank the authors for responding to my comments. However, the current version would be more improved if the authors address the following issues, which were mentioned but haven't been resolved thoroughly:

1. As to my previous comment 2.5 ("In the legends of FIGUREs 3, 4, 9, 10, and 13, it would be more rigorous to mention BOTH the sample size (the number of data points) AND whether the data points (in FIGURE 13) were technical or biological replicates"), the authors did not seem to supplement these figures' legends with the sample size. This revision would render the current study more rigorous, so please mention the sample size in the figure legends.

Experimental design

N/A

Validity of the findings

N/A

Additional comments

N/A

---

## Round 0.3 · accepted · Accept

I am happy with this revised version.

Reviewer 3 ·

Basic reporting

Thank the authors for responding to all of the comments. The current version has been much improved.

Experimental design

N/A

Validity of the findings

N/A

Additional comments

N/A